# Identification of Novel Covalent XPO1 Inhibitors Based on a Hybrid Virtual Screening Strategy

**DOI:** 10.3390/molecules27082543

**Published:** 2022-04-14

**Authors:** Zheyuan Shen, Weihao Zhuang, Kang Li, Yu Guo, Bingxue Qu, Sikang Chen, Jian Gao, Jing Liu, Lei Xu, Xiaowu Dong, Jinxin Che, Qimeng Li

**Affiliations:** 1Department of Urology, Rui’an People’s Hospital, the Third Affiliated Hospital of Wenzhou Medical University, Wenzhou 325200, China; 182800@zju.edu.cn; 2Innovation Institute for Artificial Intelligence in Medicine of Zhejiang University, Hangzhou 310018, China; 22019051@zju.edu.cn (S.C.); 21919039@zju.edu.cn (J.G.); dongxw@zju.edu.cn (X.D.); 3Hangzhou Institute of Innovative Medicine, Institute of Drug Discovery and Design, College of Pharmaceutical Sciences, Zhejiang University, Hangzhou 310058, China; weihaoz0920985@zju.edu.cn (W.Z.); yu.guo@zju.edu.cn (Y.G.); bingxuequ@zju.edu.cn (B.Q.); iami8888@163.com (J.L.); 4Jiangsu Hengrui Pharmaceuticals Co., Ltd., Shanghai 222000, China; kang.li@hengrui.com; 5Institute of Bioinformatics and Medical Engineering, School of Electrical and Information Engineering, Jiangsu University of Technology, Changzhou 213001, China; tianzhiya830118@163.com

**Keywords:** nuclear export protein 1, hybrid virtual screening, covalent docking, anti-tumor

## Abstract

Nuclear export protein 1 (XPO1), a member of the nuclear export protein-p (Karyopherin-P) superfamily, regulates the transport of “cargo” proteins. To facilitate this important process, which is essential for cellular homeostasis, XPO1 must first recognize and bind the cargo proteins. To inhibit this process, small molecule inhibitors have been designed that inhibit XPO1 activity through covalent binding. However, the scaffolds for these inhibitors are very limited. While virtual screening may be used to expand the diversity of the XPO1 inhibitor skeleton, enormous computational resources would be required to accomplish this using traditional screening methods. In the present study, we report the development of a hybrid virtual screening workflow and its application in XPO1 covalent inhibitor screening. After screening, several promising XPO1 covalent molecules were obtained. Of these, compound **8** performed well in both tumor cell proliferation assays and a nuclear export inhibition assay. In addition, molecular dynamics simulations were performed to provide information on the mode of interaction of compound **8** with XPO1. This research has identified a promising new scaffold for XPO1 inhibitors, and it demonstrates an effective and resource-saving workflow for identifying new covalent inhibitors.

## 1. Introduction

Nuclear export protein 1 (XPO1), also known as chromosomal region maintenance 1 (CRM1) [1], is a ubiquitous eukaryotic protein that mediates the nuclear export of various proteins and RNAs [2]. As a member of the nuclear export protein-p (Karyopherin-P) superfamily [3], XPO1 recognizes and binds to “cargo” proteins and transports them across the nuclear membrane. This normal cellular transport process is accomplished via the formation of an intermediate state with Ran-GTP (the nuclear transport ternary complex), and subsequent transport of this complex through the nuclear pore [4,5]. The cargo proteins transported through the nuclear pore include tumor suppressor proteins (TSPs), growth regulatory proteins (GRPs), and anti-apoptotic proteins [6,7,8].

In various blood cancers and solid tumors, XPO1 is observed to be overexpressed. Interestingly, selective inhibition of XPO1 results in an accumulation of TSPs and GRPs in the nucleus, and their nuclear sequestration impedes the occurrence and development of tumors. Thus, XPO1 inhibitors present a new target mechanism for the design of anti-tumor drugs. Because of the essential biological functions of XPO1, the development and characterization of inhibitors of XPO1 have both received extensive attention, and several XPO1 inhibitors are currently under clinical investigation. In general, small molecule inhibitors of XPO1 prevent the formation of the nuclear transport ternary complex by covalently binding to Cys 539 of XPO1 [9] (also reported as Cys 528 in some research articles) [10].

Leptomycin (LMB) was the first XPO1 inhibitor identified to bind completely and irreversibly to XPO1 [11]. Due to its severe systemic toxicity, LMB was eventually replaced by other small molecule inhibitors [12,13]. Selinexor was approved as the world’s first orally administered selective inhibitor of nuclear export (SINE) [14]. This inhibitor slowly and reversibly binds to XPO1, causing TSPs to accumulate in the nucleus. As a consequence, tumor suppressor function is reset and amplified. While this leads to apoptosis in cancer cells, normal cells are not affected [15]. Eltanexor, a second-generation SINE molecule, is currently in clinical studies. In preclinical models, Eltanexor exhibited similar in vitro efficacy to Selinexor, with the added advantages of a wide therapeutic window, low blood-brain barrier penetration, and high tolerability [16,17,18]. In addition to Eltanexor (KPT-8602), several other XPO1 inhibitors are currently under investigation worldwide, including Verdinexor (KPT-335), BIIB-100 (KPT-185/KPT-350), KPT-127, KPT-214, KPT-251, and KPT-276. However, all of these inhibitors have extremely similar structures (Figure 1). Therefore, the exploration of new covalent inhibitor backbones for XPO1 is of great significance [19,20].

The virtual screening (VS) approach has been widely used in drug discovery, and it is a powerful method for identifying new active scaffolds [21]. However, relatively few virtual screening studies have been performed using covalent docking. In particular, covalent docking is known to be computationally intensive, and few research institutions have the computing power to support a high-performance virtual screening from a large library of covalent compounds [22,23,24,25,26,27]. Expanding on our previous research [28,29,30,31,32,33,34,35,36], we report in this study the development of a hybrid screening tool that combines non-covalent docking, covalent docking, and pose filters to improve the discovery efficiency for covalent molecules. While covalent virtual screening of 10,000 molecular libraries with the same 48-core computer configuration would take 416 days using traditional methods, the hybrid screening method required only 10 days. Using our novel hybrid screening method, we were able to perform covalent virtual screening of large compound libraries with limited computational resources while improving the hit rate of lead compound discovery.

We applied our hybrid workflow method to the virtual screening of XPO1 covalent inhibitors. First, non-covalent docking software and protein models were used to screen a self-constructed covalent compound library. Next, molecules with a covalent target in proximity to Cys 539 were rapidly filtered using a pose filter, which is the key to coupling covalent and non-covalent steps to promote screening efficiency. Molecules obtained from pose filtering were used as input for covalent docking. After a covalent docking process, a series of novel compound skeletons with potential XPO1 inhibitor activity was discovered. We then performed a cell-based XPO1 inhibitory activity assay on the discovered compounds to evaluate their inhibitory effect. Using this method, we identified XPO1 inhibitors with moderate inhibitory activity that had significantly different backbone types from the existing XPO1 inhibitors. These exciting results demonstrate that our workflow constitutes a promising protocol for covalent virtual screening. Moreover, our workflow provides an essential structural and experimental basis for discovering XPO1 inhibitors with novel structures and high activity.

## 2. Results

### 2.1. Evaluation of Screening Method

To identify solutions capable of enriching XPO1 high-affinity compounds, we evaluated a range of software and protein structures. The high-resolution protein structures were all obtained from the RCSB protein data bank. Due to the low resolution of the crystal structure of full-length Homo sapiens XPO1 (both above 2.5 Å), we used the protein structure of Saccharomyces cerevisiae in selecting the structure. The sequences of Saccharomyces cerevisiae and Homo sapiens XPO1 have an overall 47% similarity and the structure of the binding site pocket is homologous. The following series of human derived XPO1 protein structures were evaluated in our efforts to determine the optimal experimental conditions: 5UWH; 5UWI; 5UWJ; 5UWO; 5UWP; 5UWQ; 5UWR; 5UWS; 5UWT; 5UWU; 5UWW and 6CIT [37,38].

To facilitate docking, the XPO1 structure was modified with an amino acid mutation (C539G) that eliminated steric interference under non-covalent conditions. The protein structure was then prepared using Schrödinger’s protein preparation wizard module. The prepared protein structures were evaluated with LeDock and AutoDock software to determine whether these experimental conditions could effectively distinguish between inhibitors and decoys. In total, seventy effective XPO1 inhibitors and 5000 random decoys were evaluated as ligands docked to the mutation pocket. During the docking process, the docking scores were collected to calculate *p*-values. Using the *p*-values and the area under the ROC curve, the best performances were obtained using the 5UWI and 5UWO structures with the LeDock software. According to the Frequency histogram, 5UWO exhibited excellent discrimination and was chosen for use in the subsequent screening process (Figure 2).

### 2.2. Hybrid Virtual Screening Process for XPO1 Covalent Inhibitor Discovery

The virtual screening protocol employed to discover XPO1 inhibitors is illustrated in Figure 3a. We first established a Michael compound library. After a first round of non-covalent screening of the Michael compound library, 8986 compounds were selected and prepared for the next covalent round. After redocking with Glide_SP and the pose filter, the selected compounds were further filtered using a fast covalent docking round and a thorough covalent docking round. As a result of this screening process, nine molecules with reasonable conformations and high scores were manually selected to enrich skeleton diversity (Table 1). The molecular skeletons of these molecules should provide a reference for subsequent drug design (Figure 3b).

### 2.3. Compound **8** Shows Promising Anti-Proliferation Activity

Since there is no protein-based evaluation model for XPO1, researchers commonly use a model based on MM.1S survival to evaluate the activity of XPO1 inhibitors [39]. The selected compounds were subjected to an anti-proliferation test using MM.1S cells to confirm the activities of the identified compounds. Compound **1**, compound **2**, and compound **8** exhibited significant inhibitory effects on the proliferation of MM.1S cells (Figure 4a). Furthermore, compound **1** and compound **8** exhibited ~90% inhibition at concentrations of 10 μM. After further scrutiny, it was decided that compound **8** exhibited more potential for exploitation. Therefore, we subsequently tested compound **8** on several other cell lines (A549, PC3, MDA-MB-231) to evaluate potential off-target problems. As shown in the figure below, no inhibitory effect of compound **8** was observed in these cellular assays (Figure 4b).

In an earlier study, the Stewart group reported that knockdown of XPO1 had no effect on the survival of lung cancer (A549) cells, which indicated that the cellular fate of A549 cells was independent of the function of XPO1 [40]. Thus, inhibition of XPO1 should not affect the survival of A549 cells. Our experimental result reported here using compound **8** on A549 cells confirms this observation.

### 2.4. Compound **8** Shows a Moderate Inhibitory Effect on Nuclear Export

Next, we performed cellular immunofluorescence assays on MM.1S cells using confocal microscopy to provide direct (visual) evidence of the interaction of compound **8** with XPO1. Ran binding protein 1 (RanBP1), a nuclear-cytoplasmic shuttle protein localized to the cell membrane of interphase cells, is known to play an important role in nuclear transport. Because RanBP1 contains an NES near its C-terminus, it is a substrate for XPO1. As a consequence, shuttling of RanBP1 between the nucleus and the cytoplasm is dependent on XPO1 activity. Indeed, Macara et al. demonstrated that RanBP1 accumulates in the nucleus after treatment with the XPO1 inhibitor Leptomycin B [41]. Thus, the location of RanBP1 can be used to report on XPO1 activity after treatment with an export inhibitor. In control assays (DMSO only), RanBP1 was detected mainly in the merging channels around the cytoplasm and nucleus. However, when the cells were treated with compound **8**, the green fluorescent spots and the blue fluorescent spots largely overlapped when viewed through the merged channels. Notably, the results obtained with compound **8** were the same as those obtained with KPT-330, indicating that compound **8** inhibited the activity of XPO1 directly (Figure 5).

### 2.5. Molecular Dynamics Investigation

We also performed a 100 ns molecular dynamics simulation using complexes of XPO1 and compound **8**. These complexes were obtained after the final wave of covalent docking. An analysis of the results reveals that the system reached equilibrium after 10 ns, and the root-mean-squared deviation (RMSD) was in the range of 1 Å. Thus, a 100 ns molecular dynamics simulation was sufficient to provide a relatively reliable final conformation (Figure 6a).

An analysis of the binding mode of compound **8** is presented in Figure 6b–d. Compound **8** binds to XPO1 in three main ways. Firstly, compound **8** covalently binds to Cys 539 (as determined in previous docking studies), and this interaction constitutes a localization effect. Secondly, a π-π interaction is formed between Lys 579 and the furan ring (and this extends to the protein interior). Thirdly, the terminal chlorine substituted benzene ring is involved in π-cation and π-π interactions with Lys 548 and Phe 583.

## 3. Discussion

To overcome the high computational demands of covalent virtual screening and facilitate the search for new backbone inhibitors of XPO1, we established a new virtual screening workflow for covalent compounds. This was accomplished by combining different docking methods in such a way as to accelerate significantly the screening speed for covalent compounds (while maintaining screening reliability). We applied this workflow to the screening of XPO1 covalent compounds and identified three promising novel XPO1 covalent inhibitor molecules. The three discovered molecules—compound **1**, compound **2**, and compound **8**—all adopted different parent cores: 3-(benzo[d][1,3]dioxol-5-ylmethyl)-2-thioxothiazolidin-4-one; (E)-2-((3-chlorophenyl)imino)thiazolidine-4-one and 2-(3-(trifluoromethyl)phenyl)furan. These backbones are markedly different from those of currently developed XPO1 inhibitors, which are based on acrylates or acrylamides and multi-substituted benzene ring substituted triazoles. Of the three discovered compounds, compound **8** exhibited high activity in both inhibition of proliferation assays and a nuclear export inhibition assay (with an IC_50_ of 4443 nM in hematological tumor cell lines). Our in-silico research also provides the molecular binding mechanism of compound **8**, providing a reference for the subsequent modification of compound **8**, and the expansion of XPO1-related compounds. Further optimization of compound **8** is currently ongoing in our laboratory.

## 4. Materials and Methods

### 4.1. Covalent Compound Library Production

Many compound libraries are available for virtual screening, but high-quality “warhead” covalent small molecule libraries are scarce. Commercial libraries of covalent compounds established by reagent suppliers (3405 molecules) typically contain small volumes of small molecules. In many cases, the reactive potential of the small molecules is unproven, which seriously affects the accuracy and efficiency of virtual screening. To address this problem, we first established a high-quality, high-volume compound library with one reactive warhead for Michael acceptors (124,721 molecules). To ensure the molecular diversity of our covalent compound library, compounds from ChemDiv compound libraries and TargetMol with well-developed skeleton diversities were initially included. The filtering process for compounds containing Michael acceptors was then implemented using the Structure Filtering module in Schrödinger 2020-3 software. Using the conformational restriction of +[C,c] = [C,c] − [C,c,S,s] = [O] substructure, a final covalent small molecule library of 124,721 Michael acceptors was built.

### 4.2. Protein and Software Evaluation

The inhibitor and decoy data were obtained from a previous article [26]. The XPO1 protein structures were docked with various inhibitors and decoys using several software suites. The scoring data obtained from individual docking procedures were recorded separately. We then analyzed and evaluated the agreement between activity and scores using various evaluation procedures, including a double *t*-test, ROC curve, and frequency histogram.

### 4.3. Non-Covalent and Covalent-Coupling Workflow

To avoid steric clashes at the active site, we initially mutated Cys 539, which undergoes the covalent reaction, to Gly539. This mutated 5UWO molecular structure was used with LeDock software for the first round of non-covalent screening of the Michael compound library. After filtering, 8986 compounds with binding energies ≤ −6.5 kcal were selected, and these were prepared for further docking using the ligprep module. The prepared molecules were then docked using the Glide_SP function to output more binding conformations. Next, the Pose filter function in Schrödinger software was used to filter out docking conformations that could not form a covalent bond with XPO1. The restriction in the Pose filter process was that the Michael addition donor substructure in the ligand must be within a 5 Å radius of Gly 539. Following this filtering process, the remaining 3018 molecules were covalently screened using the Fast-Docking function in Covalent Docking. The unmutated structure containing Cys 539 was used for this covalent docking procedure (and all subsequent procedures). After covalent docking screening, 64 small molecules with a score ≤ −6.8 were selected. Finally, the Thorough-Docking module was used to perform conformationally exhaustive covalent docking to facilitate optimal docking of the selected small molecules with XPO1.

### 4.4. Molecular Dynamics

We used the optimal conformation of the compound **8**-XPO1 complex (with the top score in the covalent docking procedure) as an input for covalent molecular dynamics. First, a TIP3P aqueous box was constructed for the optimal complex conformation using the System Builder module. The volume of the aqueous box was then minimized, and the system was neutralized with salt (force field, OPLS3e; remaining options, default parameters). The molecular dynamics simulations were performed using the Molecular Dynamics module at NPT (300.0 K, 1.01325 bar) for 100 ns, and trajectories were recorded at 100 ps intervals. The RMSD values, non-covalent interactions, and covalent interactions of proteins and ligands in the kinetic trajectories were subsequently analyzed using the Simulation Interactions Diagram module.

### 4.5. Cell Proliferation Assays

The inhibitory effects of nine test compounds on the proliferation ability of MM.1S cells in vitro were examined using the CCK-8 kit method. To thoroughly investigate the inhibitory activity of compound **8** against XPO1, we tested its effect on the proliferation ability of a diverse range of cell lines. The four cell lines used were: A549 cells, which are adherent non-small cell lung cancer cells cultured in RPIM-1640 medium; MM.1S cells, which are semi-adherent multiple myeloma cells cultured in RPIM-1640 medium; PC3 cells, which are adherent prostate cancer cells cultured in RPIM-1640 medium and MDA-MB-231 cells, which are adherent breast cancer cells cultured in DMEM medium. After initial culturing, the cell lines were seeded separately in 96-well plates for subsequent testing. Compound **8** (10 µM) was serially diluted (3-fold each time) to yield a serial dilution series of nine different concentrations. The cells (in individual wells) were then treated with different concentrations of compound **8** for 48 h. Finally, 10 µL of CCK8 assay reagent was added, the cells were further incubated for 1 h at 37 °C, and the OD was detected at 450 nm. The cell viability was plotted on GraphPad Prism 7.

### 4.6. Immunofluorescence Staining of RanBP1

KPT-330 (100 nM) was used as a positive control. An assay solution containing no Compound **8** (0 μM) was used as a negative control. Compound **8** was tested at three different concentrations (1 μM, 10 μM, and 20 μM). A semi-adherent culture of MM.1S cells was used for all testing. After resuscitation, MM.1S cells were expanded in culture and then seeded into 6-well plates. The cells in separate wells were then incubated with one of the two controls or one of the three concentrations of compound **8**. After 24 h, the treated cells were harvested, washed by PBS centrifugation, and fixed with 4% paraformaldehyde. Triton X-100 was added to permeabilize the cells, and the cells were then blocked with 4% BSA. After processing for primary antibody binding and secondary antibody binding, the cells were treated with Hoechst nuclear stain. Finally, the cells were imaged using confocal microscopy. The resulting image data were subsequently analyzed and processed.

## Figures and Tables

**Figure 1 molecules-27-02543-f001:**
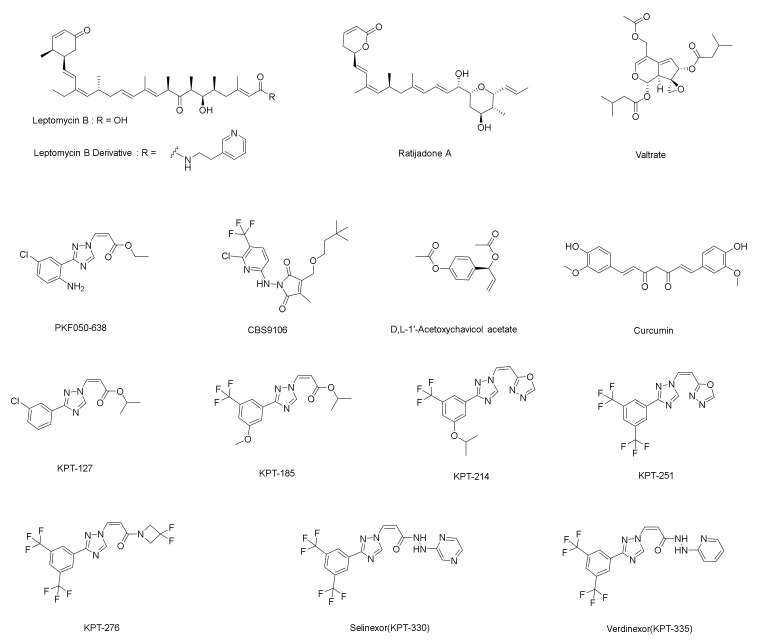
The principal XPO1 inhibitors currently under investigation worldwide.

**Figure 2 molecules-27-02543-f002:**
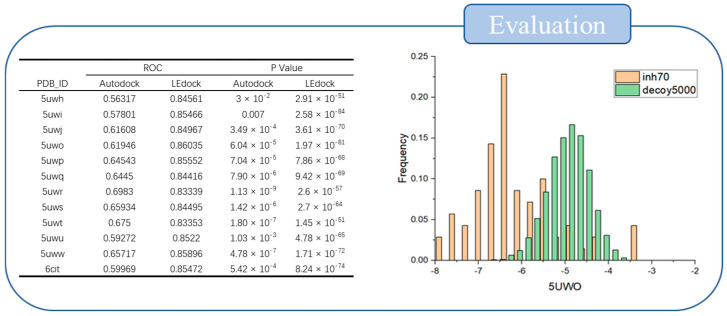
The evaluation results reveal that optimal screening proficiency was obtained using the 5UWO structure with LeDock software.

**Figure 3 molecules-27-02543-f003:**
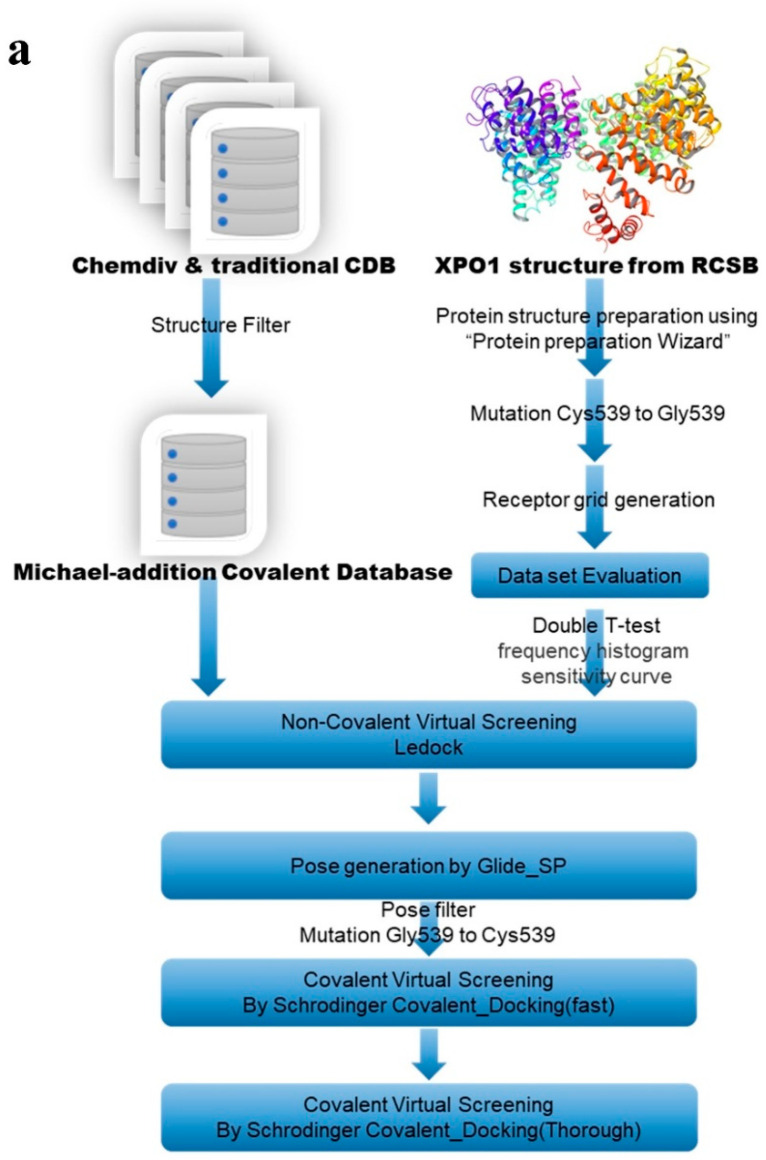
A description of our workflow and descriptions of the selected molecules. (**a**) Hybrid workflow coupling non-covalent and covalent screening processes for XPO1. (**b**) Characteristics of the final selected molecules.

**Figure 4 molecules-27-02543-f004:**
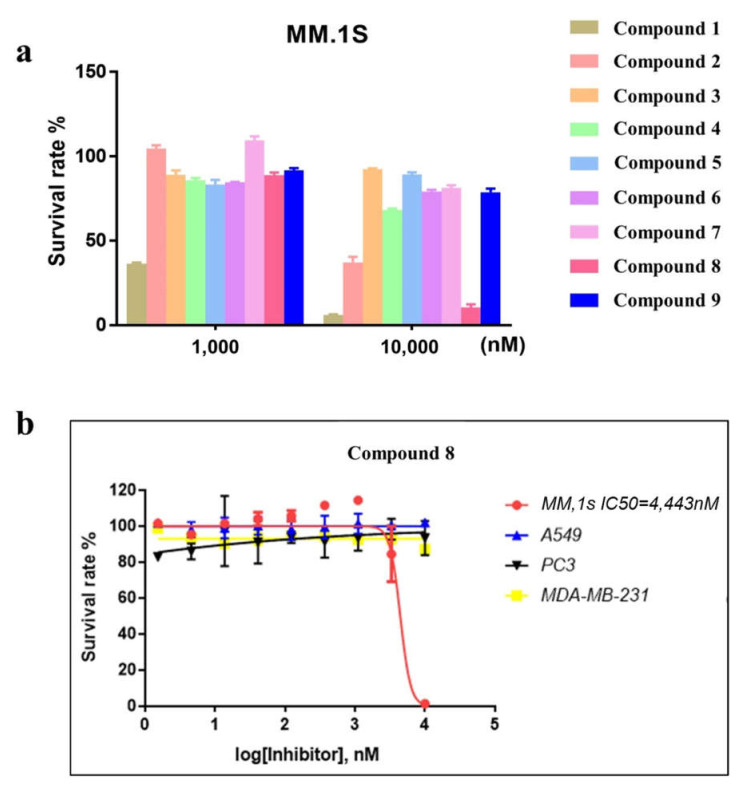
(**a**) Compounds **1**, **8**, and **9** exhibited optimal XPO1 inhibitory activity in three independent sets of cell proliferation assays. (**b**) Inhibition of proliferation in MM.1S, A549, MDA-MB-231, and PC3 cell lines by compound **8** treatment after 48 h.

**Figure 5 molecules-27-02543-f005:**
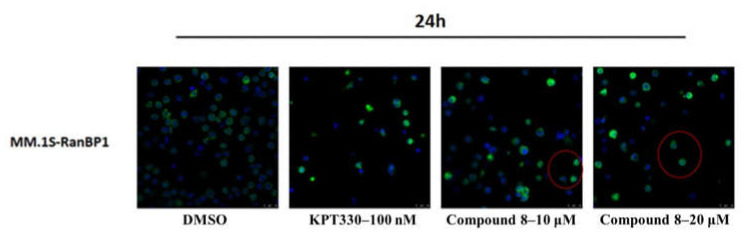
Compound **8** demonstrated moderate nuclear export inhibition in the nuclear export inhibition experiment.

**Figure 6 molecules-27-02543-f006:**
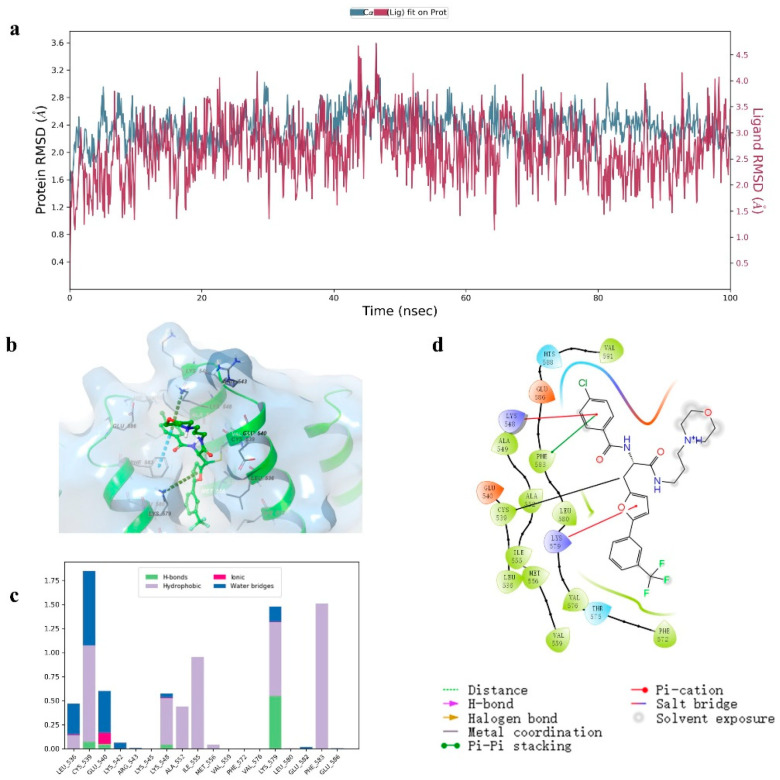
Compound **8**-XPO1 complex molecular dynamics result. (**a**) RMSD of proteins and small molecules. (**b**) Global view of compound **8**-XPO1 interactions. (**c**) Contribution of each amino acid residue to the binding affinity. (**d**) 2D sketch showing binding pocket detail.

**Table 1 molecules-27-02543-t001:** Docking score data for selected small molecules.

Entry	Title	Force Field	Prime Energy Solvent Model	RXN	Docking Score	MM-GBSA dG Bind
Compound **1**	4578–1143	OPLS3e	VSGB2.1	Michael Addition	−7.333	−59.68
Compound **2**	5090–1053	OPLS3e	VSGB2.1	Michael Addition	−7.314	−58.60
Compound **3**	Y600–2961	OPLS3e	VSGB2.1	Michael Addition	−7.493	−59.50
Compound **4**	K291–0098	OPLS3e	VSGB2.1	Michael Addition	−7.116	−51.65
Compound **5**	V029–9645	OPLS3e	VSGB2.1	Michael Addition	−6.902	−69.82
Compound **6**	V003–8338	OPLS3e	VSGB2.1	Michael Addition	−7.451	−59.33
Compound **7**	V025–8829	OPLS3e	VSGB2.1	Michael Addition	−8.072	−60.33
compound **8**	4476–4961	OPLS3e	VSGB2.1	Michael Addition	−7.164	−61.77
Compound **9**	2733–3746	OPLS3e	VSGB2.1	Michael Addition	−7.135	−63.64

Force fields, solvent models, covalent reactions, docking scoring, MM/GBSA are all listed in the table.

## Data Availability

Not applicable.

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
