# Peer review of "Identification of Novel Covalent XPO1 Inhibitors Based on a Hybrid Virtual Screening Strategy"

_molecules, 2022, doi:10.3390/molecules27082543_

Round 1

Reviewer 1 Report

This article may be proceed further

Author Response

Comments and Suggestions for Authors:

This article may be proceed further

Response:Thank you very much for your suggestions and interest in our results. The optimization of the new skeleton inhibitor of XPO1 that we have discovered is currently in progress. In parallel, the hybrid screening workflow that we have built will also be applied in the discovery process of other covalent inhibitors.

Reviewer 2 Report

The authors have replied to my different comments.

Author Response

Comments and Suggestions for Authors:

The authors have replied to my different comments.

Response: Thank you for your suggestion. All of these comments are of great importance to improve the quality of our article. 

Reviewer 3 Report

The authors have submitted a revised manuscript of the paper entitled “Identification of novel covalent XPO1 inhibitors based on a hybrid virtual screening strategy.” I am glad that the authors followed my recommendations and really improved the text. The language is easily understandable and it is possible to focus on the facts. As I have already mentioned, the topic is interesting and well processed. Instead of laborious and computational time consuming process of seeking for a potential covalent binder of nuclear export protein 1 (XPO1), the authors utilized less demanding combination of noncovalent and covalent inhibitors to find out a potential covalent inhibitor. I believe that such process can save a lot of computational time and at the end it is more efficient. Nevertheless, I have a couple of notes and objections.

l. 89: something is missing here, probably the word effect (inhibitory effect)

l. 195, 197: I don’t understand what kind of π-π interaction could be formed by lysine residue? It is hard to check it in figure because the quality of figure 6b and 6d remained really very low. And it is related to my final objection. The resolution of both figures is absolutely not sufficient and it should be improved.

Author Response

Comments and Suggestions for Authors:

l. 89: something is missing here, probably the word effect (inhibitory effect)

Response:  Thanks for pointing out this issue. We are sorry for our carelessness.  We have made the corrections and highlighted the changes based on your comments.

Ⅱ. 195, 197: I don’t understand what kind of π-π interaction could be formed by lysine residue? It is hard to check it in figure because the quality of figure 6b and 6d remained really very low. And it is related to my final objection. The resolution of both figures is absolutely not sufficient and it should be improved.

Response:  Thanks for pointing out this issue. We apologize that the formatting of the manuscript changed during the upload process, resulting in a drop in resolution that impacted the article's readability. Figure 6 has been attached. The issue of π-π interactions resulting from lysine is due to a writing mistake. The interaction force of lysine with chlorine substituted benzene ring is π-cation. We have made the corrections and highlighted the change. 

This manuscript is a resubmission of an earlier submission. The following is a list of the peer review reports and author responses from that submission.

Round 1

Reviewer 1 Report

Article entitled “Identification of Novel Covalent Xpo1 Inhibitors Based on a Hybrid Virtual Screening Strategy” article looks interesting and may be considered after revision

  1. Authors need to justify the significance of their hybrid method
  2. How this method helps in limitation in the drug discovery, discuss by comparing other methods.
  3. Line no 60: the top 9 compounds ,screened with this hybrid method are not have the same pattern, not (much) similar.
  4. Line no 184 and 186: small volumes? high volumes? mention exact number
  5. Have you compared the interaction before and after MD simulation ?
  6. Add hydrogen bond plot figure and discuss the same .
  7. Line no 90: …determine the appropriate screening methods and protein structures? On what basis?

Reviewer 2 Report

The authors report a hybrid virtual screening approach to identify novel covalent Xpo1 Inhibitors.

They first identified the best high resolution protein structure to perform their in silico protocol. They prepared a set of 70 known Xpo1 inhibitors and 5000 decoys as negative set. They evaluated the docking performance and selected 5UWO, as the best target to discriminate between true inhibitors and decoys.

They then prepared a virtual chemical library of 124,721 reactive compounds containing Michael acceptors as electrophiles.

This library was used to perform non covalent docking on the selected protein structure (mutant C539G) leading to the selection of 8986 compounds. New poses were generated for these compounds with Glide and compounds that cannot form covalent bonds, because of the distance between the Michael acceptor and residue 539, were discarded. The remaining 3018 molecules were subjected to covalent docking with the wild-type structure (C539) in a two-step process (fast and thorough). In the end, 9 compounds were retained as putative covalent inhibitors.

These compounds were tested experimentally showing that 3 compounds exhibit significant inhibitory effects on the proliferation of MM.1S cells at 10uM. The most promising compound exhibited an IC50 of approximatively 5uM in nuclear export inhibition assay

Finally, they performed a 100ns molecular dynamics simulation of XPO1 with compound 8 to assess the binding conformation of this compound.

General comments

Overall, this study is sound and convincing, however, the presentation could be greatly improved especially the figures and material and methods parts. Methods should be described in more details in figure legends and in the experimental part.

For example, it is not clear if the MD simulation was performed on the reversible compound 8 (and mutant C539G) or with the covalent form. Which PDB structure was used ? P9 l158, the authors state that compound 8 reacted with Cys539. The formation of the covalent bond cannot be observed during the MD process (unless they used a non-standard MD protocol which should be described in more details).

Why the authors did not use Human Xpo1  structures ? What is the percentage of identity between Human and Yeast sequences ?

Residue C539 is located in an alpha helix. The mutation to glycine could destabilize the α-helix, why the author did not use a mutation to alanine, which is more common (to preserve the β carbon)?

Covalent inhibition is a two‐step process. First, an inhibitor reversibly associates with the target, thus bringing the electrophilic warhead of the inhibitor within close proximity of a reactive nucleophilic residue in the targeted protein. In the second step, reaction occurs between the two reactive entities to form a covalent bond. The recognition step is driven by the binding affinity of the compound and can be assessed with docking (as performed here). The second step is driven by the chemical reactivities of the nucleophile and electrophile. How did the author assess the reactivity of the compounds during the covalent docking ? What is the solvent accessibility of C539 ?

The figures could be greatly improved, in general they are very small and the quality is poor (2D structures in Figure 3b or 3D structures in Figure 6b). What is the meaning of the different colors in figure 6c ? It is impossible to read the name of the amino acids in Fig 6c and 6d.

Minor points :

P2 l69-70, part of the phrase is missing.

Reviewer 3 Report

The authors present a manuscript of paper concerning the identification of novel covalent inhibitors of the nuclear export protein XPO1. They also show a new virtual strategy of screening the database of compounds to find out active molecules. First of all I have to say that the topic might be interesting for the readers of Molecules. However, the manuscript suffers by so many problems starting with very bad English which leads in many cases to the inability of a reader to understand the text. Furthermore, the authors did not pay attention to careful assembly of their results which sometimes leads to a total chaos. The manuscript contains technical expressions without any previous explanations. Many paragraphs lack logical connections with already written text. The quality of the figures is very low.

By my opinion, it is useless to start listing all single errors, I strongly recommend the authors to rewrite the whole manuscript, to let the text be read by someone, who is more proficient in English and after that to submit it once again.